# Investigation of the Performance of TDoA-Based Localization Over LoRaWAN in Theory and Practice

**DOI:** 10.3390/s20195464

**Published:** 2020-09-23

**Authors:** Jan Pospisil, Radek Fujdiak, Konstantin Mikhaylov

**Affiliations:** 1Department of Telecommunications, Brno University of Technology, Technicka 12, 616 00 Brno, Czech Republic; fujdiak@feec.vutbr.cz; 2Centre for Wireless Communications, University of Oulu, Erkki Koiso-Kanttilan katu 3, 90014 Oulu, Finland; konstantin.mikhaylov@oulu.fi

**Keywords:** TDoA, geolocation, localization, LoRaWAN, LPWAN, LoRa

## Abstract

The paper deals with the localization in a low-power wide-area-network (LPWAN) operating long-range wide-area-network (LoRaWAN) technology. The LoRaWAN is, today, one of the most widely used connectivity-enabling technologies for the battery-powered smart devices employed in a broad range of applications. Many of these applications either require or can benefit from the availability of geolocation information. The use of global positioning system (GPS) technology is restrained by the bad propagation of the signal when the device is hidden indoors, and by energy consumption such a receiver would require. Therefore, this paper focuses on an alternative solution implying the use of the information readily available in the LoRaWAN network and application of the time difference of arrival (TDoA) method for the passive geolocation of end-devices in the network. First, the limits of geolocation services in networks that use narrow-band communication channels are discussed, as well as the relevant challenges faced by the TDoA approach. Then, we select five classic TDoA algorithms and evaluate their performance using simulation. Based on these results, we select the two providing the best accuracy (i.e., Chan’s and Foy’s). These algorithms were tested by the field measurements, using the specially designed low-cost gateways and test devices to estimate their real-life performance.

## 1. Introduction

With the increase of interest in the internet of things (IoT) devices, which aim to simplify people’s daily routines, diverse requirements such as required connectivity throughput and power supply arise. Since the IoT concept covers devices with maximum data rates ranging from units of bits to tens of megabits per second, communication distance and device lifetime may vary as well [1]. The widely used short-range radio technologies (e.g., Wi-Fi and Bluetooth) are not adapted to scenarios that require transmission over long distances (hundreds of meters or more). Cellular communication solutions (mobile networks—3G, 4G, and others) can provide greater coverage but are very energy-consuming. The requirements of the newly developed IoT applications have led to the emergence of a new wireless technology class—low-power wide-area-networks (LPWANs) are low-power (LP) networks with extensive geographic coverage (WAN) characterized by a low bit rate. Devices that implement the protocols, that meet these requirements, also often have to be inexpensive and feature a long battery lifetime [2].

The long-range wide-area-network (LoRaWAN) is one of the most widely used LPWA technologies today and is an open standard created by the LoRa Alliance [3], which defines the communication protocol and network architecture (OSI Layer 2—Data Link Layer). This protocol is built on top of the proprietary modulation called LoRa (OSI Layer 1—Physical Layer). This modulation is used to establish a wireless communication link over a distance of tens of kilometers outdoors at data rates up to dozens of kilobits per second while operating with the maximum transmit power of only 14 dBm (in Europe) [4]. LoRaWAN technology is suitable for providing connectivity to battery-powered smart devices that will find use in a broad range of applications such as monitoring, logistics, transportation, and automation. Another feature that makes LoRaWAN especially well-suited for localization is its cell-free nature. Unlike conventional cellular systems, LoRaWAN end-devices are not tied to a specific gateway (GW). Instead, each LoRaWAN GW receives the uplink packets sent by each device and forwards them to a single network server, which processes them and removes the duplicates.

One of the possible added values of LPWAN technologies, including the LoRaWAN, is the provision of end-device positioning services. Although the global navigation satellite system (GNSS), and, specifically, the global positioning system (GPS) offers accurate location and navigation services, this imposes the limitation coming with the poor indoor propagation of the GPS signal [5]. Among the other overheads that an integration of GNSS into the IoT devices imposes are the increase of the monetary costs and energy consumption. Notably, the energy consumption raises both due to the actual positioning and the need of transferring these data wirelessly. Finally, it is worth noting that a number of the LoRaWAN devices have already been deployed with no GNSS chips on board. For this reason, a backward-compatible solution allowing us to localize the IoT devices in the network with no energy, cost, or latency overheads and without using any additional components is highly desirable. One of the perspective approaches to address this challenge, the utility and performance of which we investigate in this study, is to use time difference of arrival (TDoA) method for passive geolocation of the end-devices in the network based on the information already available in a LoRaWAN network. Importantly, this approach does not impose any hardware or firmware changes for the end devices and thus can even enable localization of the devices already deployed in-field.

### 1.1. Our Contribution

The key contributions of this paper are:The definition and discussion of the relevant aspects of TDoA-based localization in LoRaWANs.The simulation results demonstrate the performance of the five classical TDoA-based localization algorithms.The recipe of the practical implementation and validation of a Raspberry PI based low-cost TDoA-localization enabled LoRaWAN gateway (GW).The results of the practical measurement campaign demonstrate real-life localization accuracy and performance for the two selected TDoA algorithms and a way to improve it through filtering.

The paper is organized as follows. In Section 2, possible ways of geolocation in the LoRaWAN and the results of some earlier studies in this field are discussed. Section 3 further describes the key issues and challenges related to the multipath signal propagation theory, signal processing, and the TDoA geolocation in general. The following Section 4 describes the details of TDoA geolocation algorithm approaches and compares five selected algorithms by simulation. Section 5 reports the details of the hardware developed for our measurements. Section 6 presents the results of the initial testing of the hardware and provides the relevant details of our experimental setup. Section 7 reports and discusses our experimental results. In the two final sections, we provide a brief discussion and summarize the achievements, respectively.

### 1.2. General LoRaWAN Architecture

Before proceeding, we consider it useful to revise some of the basics of the LoRaWAN technology. A LoRaWAN network is composed of four basic elements (see Figure 1):

Application server—hosts a final application that allows users to analyze and process measured data from sensors or send commands back. It also allows storing the data in a database.The network server—which is responsible for managing the network, as well as decoding and filtering duplicate packets received through GWs from end devices (and, optionally, forwarding them to a respective application server), and injecting the packets to be delivered to end-devices.One or multiple GWs—an intermediary device that forwards packets from end devices to a network server through a backbone IP network (based on, e.g., Ethernet or LTE). A network may consist of several GWs that can receive the same packet and send it to the server.One or multiple end-devices—implementing one of three LoRaWAN classes (A, B or C). Each of the classes implies a somewhat different media access procedure. The class A, implying ALOHA-like channel access with two receive windows scheduled for downlink following each uplink, is the obligatory and the most commonly used option optimized for battery-powered devices. Devices operating class B open extra receive windows at scheduled intervals. Last, class C devices continuously keep the radio in receive mode as long as they are not transmitting. Devices operating this class provide the lowest latency in downlink but are not suited to be powered by a battery. By default, LoRaWAN devices operate in the class A.

## 2. Current State of Geolocation in LoRAWAN Technology

The most widely used techniques for geolocation of wireless end-devices are based on the measurement of certain parameters (e.g., signal attenuation, signal propagation time, and angle) by other devices (e.g., the GWs) with a known location. The basic methods of geolocation include [6]:Received Signal Strength Indicator (RSSI)—operates by measuring the signal attenuation between the end-device and the GW. Being rather straightforward and cheap to implement, the typical precision of this method is rather low.Angle of Arrival (AoA)—implies using the directional antennas or antenna arrays to estimate the direction from which the GW has received the signal. This method requires a complex antenna system on each GW.Time of Arrival (ToA)—operates by estimating the propagation time of the signal between the GW and the end-device. The method requires time synchronization of GWs and end-devices. A subvariant of this method, allowing the relaxation of the synchronization requirements, is the use of a two-way communication, measuring the time from sending a request to receiving the response.Time Difference of Arrival (TDoA)—operates by measuring the difference of arrival times of the radio signal issued by an end-device and received by the different GWs. This method requires accurate GW time synchronization.

It is clear from the discussion above that not all these methods are suitable for implementation in LPWAN and hence in LoRaWAN. Considering the cost of implementing localization and the limits of the communication technologies used, only ToA/TDoA and RSSI techniques fit the LoRaWAN technology well. Both of these techniques are suitable for deployment in LPWAN networks because these are inexpensive, and there is no need to modify either hardware or firmware on the end-devices side. However, the use of the RSSI based technique is further limited by the fact that many LPWA technologies operate in the license-free bands and thus share the radio spectrum with many other radio systems. The radio signals of the latter, as well as the line-of-sight (LoS) and non-line-of-sight (NLoS) propagation, may significantly disrupt the results. The ToA technique is limited by the need for accurate synchronization between a GW and a device. For the latter, this may inquire substantial additional costs.

For these reasons, in this paper, we focus on the TDoA method. The TDoA-based method finds its shortcomings in the form of multipath signal propagation, which results for LPWANs in errors, as shown in recent literature, of up to several hundred meters [7]. Higher accuracy can be achieved in rural areas where the communication of the end-devices with the GWs takes place in a direct LoS without signal reflections. Note that if the TDoA method is used to determine the position in the LoRaWAN network, the GPS is still often needed on the GWs, as the means for clock synchronization to obtain a precise timestamp [5].

### Related Work

Some scientific studies proving LoRaWAN geolocation capabilities have already been reported. In particular, they mostly focus on the application of TDoA and RSSI techniques. It should be mentioned that none of the mentioned or proposed methods increases the overhead for packet transmission. In all cases, the transmitted data (including metadata) is stored on the server and then transferred to a separate application, which performs a passive position calculation from the stored metadata.

One of the studies performed a simulation of localization using a combination of TDoA and AoA techniques [8], but did not test these in-field. Another article compares the outdoor versus indoor localization using RSSI [9]. Further improvements have been suggested in [10]. In [11] the authors improved the TDoA geolocation results using the Kalman filter, but only on stationary placed devices and without further details. Another study combined TDoA and RSSI data and used the k nearest neighbors (kNN) algorithm, but mostly with LoRaWAN (V2) gateways that offer a better performance with result of 332.6 m of median [12]. A very similar approach to ours is taken in [13], where the results around 100 m are achieved, but only on stationary positioned end-devices. Satisfactory results are provided by [14] using TDoA achieving 200 m error and RSS error estimation around 1.2–2.5 km, unfortunately without further details of implementation. In the paper [15] authors reported the median estimation error more than 400 m for stationary device using (V1) GWs.

The comparison of the other localization results with respect to the mean error reported in the relevant papers, as well as some details on the implied test setup, are reported in Table 1.

Of the three studies dealing with TDoA localization for LoRaWAN, the whitepaper [5] provides a closer look at geolocation and measurement results but does not describe in detail the implementation, such as the algorithms used. The study [23] focuses on the localization algorithm improvement to refine the position estimate results using the mobility profiles application. The thesis [16] focuses more on the theory of localization algorithms and public network verification.

Our study, in contrast to these works, provides a comprehensive review of geolocation in a LoRaWAN network. We thoughtfully discuss the issues related to the geolocation for a narrow-band signal, the influence of noise and signal multipath, LoS vs. NLoS, the influence of geometry on the computational algorithm, and stability of time synchronization of GWs. Then, we use simulations to study the efficiency of the classical TDoA localization algorithms and select two for real-life evaluation. Finally, we conducted a set of field measurements to characterize the practical performance of TDoA in LoRaWAN.

## 3. Aspects of the Position Estimation Theory

### 3.1. Multipath Propagation

In the case of LoS, when radio waves propagate through a direct path between the transmitter and the receiver, the error in a TDoA measurement is related to the accuracy of signal detection and its timestamping, and thus can be expressed as a zero-mean Gaussian random variable. In the absence of LoS (called NLoS) between the transmitter and the receiver, the radio wave suffers from one or more multipath reflections. This leads to an extension of the traveled distance and thus of the propagation time in comparison to the LoS—see Figure 2.

Thus, the propagation time in NLoS does not correspond to the time in LoS, and a ranging error of up to several hundred meters occurs [7]. LoRa modulation is characterized by efficient communication over a multipath channel. It is thus possible to transmit data even when there are several reflections from different objects or buildings between the transmitter and the receiver. One of the major factors affecting the TDoA localization accuracy is the signal bandwidth. The minimum difference in the distance traveled by the two signals which a received can recognize is thus given by [24]
(1)$$\Delta d=\frac{c}{B}\;$$
where Δd is the distance, *c* is the speed of light, and *B* is the signal bandwidth. If a receiver receives two signals with path length differences less than Δd, i.e., a direct and a reflected signal, they will not be distinguished as separate but grouped, resulting in the sum of these signals and the addition of the phase shift. This harms the range estimation. The resulting worst-case ambiguity for the signal with 125 kHz bandwidth is $\pm \frac{\Delta d}{2}=\pm 1200$ m.

### 3.2. Signal Processing

The Cramer–Rao lower bound (CRLB) gives the lower bound of the variance of unbiased estimators of a certain parameter—if an unbiased estimator achieves this, it is considered to be fully efficient.

For TDoA, the CRLB indicates how much the ranging performance is affected by the signal bandwidth and signal-to-noise ratio (SNR) [25]:(2)$$\sigma_{r}^{2}\geq \frac{c^{2}}{4\pi^{2}B^{2}SNR}\left(1+\frac{1}{SNR}\right)\;$$
where $\sigma_{r}^{2}$ is the variance of the range estimation, *c* is the speed of light and *B* is the signal bandwidth. Figure 3 shows the CRLB for different signal bandwidths using this formula. As can be observed, the higher the signal bandwidth, the better the performance is.

Semtech’s LoRa transceiver (SX1276) can transmit signals with a bandwidth of 7.8–500 kHz, but LoRaWAN in the EU bands is typically restricted to use signals with 125 kHz bandwidth [26]. As a reference, we also plot the performance of the ultra-wideband (UWB) signals, which are known to be among the most efficient ones for localization.

### 3.3. TDoA Estimation

The distance between the *i*-th GW and the end-device is computed as
(3)$$R_{i}=\sqrt{{\left(X_{i}-x\right)}^{2}+{\left(Y_{i}-y\right)}^{2}}=\sqrt{X_{i}^{2}+Y_{i}^{2}-2X_{i}x-2Y_{i}y+x^{2}+y^{2}}\;$$
where, ($X_{i}$, $Y_{i}$) are coordinates of the *i*-th GW and (*x*, *y*) are the coordinates of the sought end-device. The range difference between the *i*-th GW relative to the first incoming (referred to as the “base” or a reference) GW is
(4)$$R_{i,1}=\Delta \tau_{i}\;\cdot \;c=R_{i}-R_{1}\;$$
where *c* is the speed of light, $R_{i}$ is the range distance between the *i*-th GW and the end-device, $R_{1}$ is the range distance between the first GW and the end-device, $\Delta \tau_{i}$ is the measured TDoA between the *i*-th GW and the first GW.

Due to the need for maximum energy and cost savings, LoRaWAN end-devices do not keep a precise clock synchronization. Since the exact moment when a device has started its transmission is unknown, the TDoA works by measuring the time difference of arrivals of the incoming message for two or more GWs. For each two GWs, a trajectory is created using the equation [27]
(5)$$\tau_{TDoA}=\tau_{1}-\tau_{2}\;$$
where $\tau_{TDoA}$ is the known difference of the GW timestamps and $\tau_{1}$ and $\tau_{2}$ are unknown. The trajectory is hyperbola-like and the target point then lies somewhere on this trajectory, as illustrated in Figure 4. This can be written as $\tau_{TDoA}=\tau_{1,1}-\tau_{1,2}=\tau_{2,1}-\tau_{2,2}=\tau_{3,1}-\tau_{3,2}=\tau_{4,1}-\tau_{4,2}.$

Typically, one of the GWs is selected as the reference GW and all the time differences from other GWs are referenced to it. This means that the number of hyperbolas (TDoAs) is always one less than the number of receiving GWs. There may occur several situations, which are dependent on the number of TDoAs. If there are less than two TDoAs (i.e., three GWs), the position cannot be determined in 2D space. If the number of TDoAs is equal to two (Figure 5a), the geolocation in 2D space is possible. The searched point is located at the intersection of the hyperbolas. If the number of TDoAs is greater than two, it is not easy to find an algebraically accurate solution, since the hyperbolas may not intersect at the same point. Then, it is necessary to use approximation techniques to find the most suitable solution (Figure 5b), or filter out some of the measurements (e.g., getting rid of the NLoS-likely ones).

## 4. Position Estimation (TDoA) Algorithms Performance and Selection

### 4.1. TDoA Geolocation Algorithms Approaches

The TDoA geolocation algorithms can be classified into two major subgroups:Non-iterative algorithms—imply the expression of unknowns by a suitable modification, e.g., a transformation of nonlinear equations to linear, and the search for one single analytical solution. The advantages of these algorithms are lower demands on computing power and known calculation time. Among the disadvantages is the lower accuracy.Iterative algorithms—repeat the measurement multiple times, attempting to refine the result with each next iteration. The criterion to stop can be finding a sufficiently accurate solution or reaching the maximum number of iterations. These algorithms enable higher positioning accuracy compared with non-iterative algorithms but may have drawbacks for their convergence, and the required computational resources and time. The choice of the initial estimate may also strongly influence the whole calculation process.

### 4.2. Comparison of Selected Algorithms for Localization in LoRaWAN

Given that for the TDoA-based localization over LoRaWAN, the GWs serve as signal receivers, their position and geometry play an important role. The positions of the GWs have been selected following the guidelines from [28]. Specifically, the GWs have been placed to form a circle around the area, where tests were carried. The map of the test area and the coordinates of the GWs are depicted in Figure 6. The deployed network covers an area of 4.58 km${}^{2}$.

**GW1:***49.387222${}^{{}^{\circ}}$ N, 15.953000${}^{{}^{\circ}}$ E* (Kochánov)**GW3:***49.398417${}^{{}^{\circ}}$ N, 15.982806${}^{{}^{\circ}}$ E* (Olší)**GW2:***49.382222${}^{{}^{\circ}}$ N, 15.977139${}^{{}^{\circ}}$ E* (Lavičky)**GW4:***49.411028${}^{{}^{\circ}}$ N, 15.951444${}^{{}^{\circ}}$ E* (Netín)

Then, we have selected five widely used algorithms for TDoA geolocation, namely Fang’s [29], Chan’s [30], spherical-intersection (SI) [31], linear-least squares (LLS) [32], and Foy’s [33]. Among these algorithms, we aim to identify the ones that provide the best accuracy of positioning.

Fang’s algorithm—non-iterative algorithm with closed-form solution, which is reduced to simple finding the suitable roots of a quadratic equation.Chan’s algorithm—provides non-iterative approach with closed-form solution and is close realization of the maximum-likelihood estimator.SI algorithm—this method is also non-iterative, which brings the closed-form solution. Unlike others, however, it calculates the intersection of spheres and not hyperboloids. The relatively small change in the eccentricity of one of the two hyperboloids can have a significant effect on the resulting position of the intersection. Thus, it suppresses the problem of hyperboloid eccentricity.LLS algorithm—provides non-iterative approach, that brings closed-form solution. This algorithm is based on least-squares (LS) method.Foy’s algorithm—this iterative approach uses the Taylor series and Taylor polynomials. The algorithm starts with an initial position estimate ($x_{0}$, $y_{0}$), which is then improved by an iterative process. Compared to non-iterative algorithms, this algorithm is more computationally complex.

For this, we randomly selected a number of points within the area bounded by the lines connecting the four GW positions. For each point, we have measured the distances to individual GWs and calculated the signal propagation time from these. To model the effect of the real environment, a random noise (ranging error) was added to the distance values expressed by a Gaussian random variable and given by
(6)$$f\left(x,\mu ,\sigma \right)=\frac{1}{\sigma \sqrt{2\pi }}e^{-\frac{1}{2}{\left(\frac{x-\mu }{\sigma }\right)}^{2}}\;,$$
where μ is the mean of the distribution and σ is the standard deviation. Then we used each of five selected algorithms to estimate the position. The estimation position error was defined as the distance between the estimated position and the original one. The error is calculated as
(7)$$\Delta d\_error=\sqrt{{\left(x-\hat{x}\right)}^{2}+{\left(y-\hat{y}\right)}^{2}}\;$$
where (*x*, *y*) are the coordinates of the original position and ($\hat{x}$, $\hat{y}$) are the estimated coordinates. Throughout the simulation μ=0 (zero mean) and $\sigma^{2}$ (variance) varied from 0 to 8000 m${}^{2}$. This variance range was divided into 27 bins with 5000 random positions checked for each bin. The resulting value of the mean absolute error (MAE) was then calculated as
(8)$$MAE=\frac{\sum_{i=1}^{n}\left|\Delta d\_error_{i}\right|}{n}$$

Of the five selected algorithms, Fang’s algorithm requires having the data from exactly three GWs. In the case if more GWs have received a packet, the selection of three GWs, based on some rule (i.e., the lowest timestamp) must be done. Further, Chan’s algorithm can be used with two or more TDoAs (i.e., 3 or 4 GWs). For the other algorithms (i.e., SI, LLS, and Foy’s), the TDoAs from at least four GWs need to be used. Figure 7 summarizes the result of the simulation.

As can be seen from the figure, the MAEs of Chan’s and Fang’s algorithms are higher than those of the other algorithms. However, unlike the other algorithms, these two used the data of only three GWs. Note, that Chan’s algorithm performed better by ≈ 10 m compared to Fang’s. Among the algorithms utilizing the data from four GWs, Chan’s, LLS, and SI are non-iterative algorithms, while Foy’s algorithm is iterative. For this reason, the latter is more computationally complex, but, as can be seen from our results, allows better localization accuracy. As reported in [34] the computation average time (for one geolocation result) comparison provided for above listed algorithms; it confirms that iterative algorithms are more computationally complex. For results in detail, see Table 2. The measurement was carried out in the GNU Octave 4.4.1, OS: macOS Catalina, RAM: 8 GB, CPU: Intel Dual-Core i5.

Based on the results of the simulations, we selected two algorithms, namely Chan’s and Foy’s, for the practical evaluation. These two were chosen since they provide the best accuracy of all the tested algorithms for the case of 2 TDoAs (3 GWs) and 3 TDoAs (4 GWs), respectively, even though they are more computationally complex. In the two following subsections, we further detail the implementation of these algorithms to be used for the real-life measurements.

### 4.3. Chan’s Algorithm

For three GWs, two TDoA hyperbolas are created, the solution is then in the form [30]
(9)$$\left[\begin{array}{c} x\\ y \end{array}\right]=-{\left[\begin{array}{cc} X_{2,1} & Y_{2,1}\\ X_{3,1} & Y_{3,1} \end{array}\right]}^{-1}\times \left\{\left[\begin{array}{c} R_{2,1}\\ R_{3,1} \end{array}\right]R_{1}+\frac{1}{2}\left[\begin{array}{c} R_{2,1}^{2}-K_{2}+K_{1}\\ R_{3,1}^{2}-K_{3}+K_{1} \end{array}\right]\right\}\;$$
where $K_{1}=X_{1}^{2}+Y_{1}^{2}$, $K_{2}=X_{2}^{2}+Y_{2}^{2}$, $K_{3}=X_{3}^{2}+Y_{3}^{2}$. *X* and *Y* denote the known positions of GWs. Subsequently $X_{i,1}$ and $Y_{i,1}$ give the distance between $GW_{i}$ and $GW_{1}$ for coordinates *x* and *y*, respectively. The values $R_{1}$ and $R_{2}$ are computed from (Equation 3). Inserting (Equation 9) into (Equation 3) with i=1 produces a quadratic equation $aR_{1}^{2}+bR_{1}^{2}+c=0$. Substitution of the positive root back into (Equation 9) gives the solution.

### 4.4. Foy’s (Taylor-Series Based) Algorithm

The algorithm requires an initial estimate ($x_{0}$, $y_{0}$). In each step, a local least-squares (LS) solution is found and then the deviation is calculated and added to the $x_{0}$ and $y_{0}$ [33]
(10)$$h=\left[\begin{array}{c} R_{2,1}-\left(R_{2}-R_{1}\right)\\ R_{3,1}-\left(R_{3}-R_{1}\right)\\ \vdots \\ R_{N,1}-\left(R_{N}-R_{1}\right) \end{array}\right]\;,\;\;\Delta =\left[\begin{array}{c} \Delta x\\ \Delta y \end{array}\right]={\left(G^{T}Q^{-1}G\right)}^{-1}G^{T}Q^{-1}h\;$$
(11)$$G=\left[\begin{array}{cc} \left(X_{1}-x\right)/R_{1}-\left(X_{2}-x\right)/R_{2} & \left(Y_{1}-y\right)/R_{1}-\left(Y_{2}-y\right)/R_{2}\\ \left(X_{3}-x\right)/R_{1}-\left(X_{3}-x\right)/R_{3} & \left(Y_{1}-y\right)/R_{1}-\left(Y_{3}-y\right)/R_{3}\\ \vdots & \vdots \\ \left(X_{N}-x\right)/R_{1}-\left(X_{N}-x\right)/R_{N} & \left(Y_{1}-y\right)/R_{1}-\left(Y_{N}-y\right)/R_{N} \end{array}\right]$$

The covariance matrix is denoted as *Q*. With each iteration Δx adds to $x_{0}$ and Δy adds to $y_{0}$. The whole process is repeated until the termination conditions are satisfied. It can be a minimal distance between $\Delta^{\prime }s$ between iterations or exceeding the maximum number of iterations.

## 5. Experimental Setup and Methods

We start the section by discussing the LoRaWAN metadata messages used as an input for the geolocation estimator and continue by detailing the hardware components designed and used by us in the measurements.

After decoding an uplink transmission from an end-device, a GW generates metadata composed of the reception timestamp, the packet’s modulation-coding scheme, the signal strength, and signal noise ratio estimates, the frequency channel, etc. These data are appended to the actual medium access control (MAC) payload and sent to the network server. The latter aggregates the messages received from the different GWs and may store these for future processing. An example of the record of the network server (captured during our real-life measurements) is provided below.



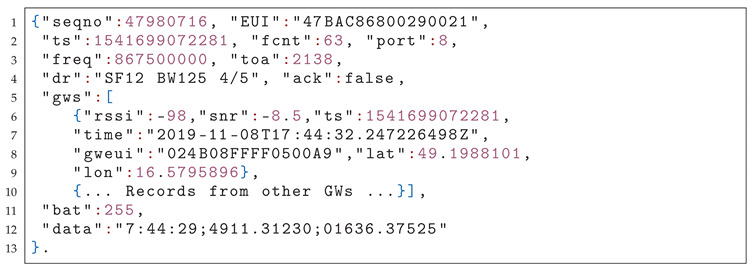



Specifically, the data record is composed of:The identifiers of the data record and the end device sending this packet (line 1).The internal network server timestamp, the sequence number of the packet, and the port identifier (line 2).The parameters of the physical layer—e.g., frequency and spreading (lines 3–4).One or more records of the GWs receiving this packet (lines 5–10). Each record includes the information about the signal and signal-to-noise ratio levels (line 6), the timestamps (lines 2 and 3), the identifier of the GW, and, if available, its location (lines 8–9).The information about the end-device battery (if available, line 11).The packet payload.

### 5.1. Gateway

As of today (2020), most commercial LoRaWAN GWs available on the market do not have a GNSS/GPS (e.g., IMST iC880A-SPI and RAK833). The GPS is featured only by several high-end products (e.g., the ones from Kerlink, Gemtech, and Link Labs) with a price of over $400. For this reason, for our experiments, we created a low-cost GPS-enabled LoRaWAN GW, composed of a Raspberry PI computer and a custom printed circuit board (PCB)—the LoRaWAN concentrator.

The principal diagram and an illustration of the manufactured concentrator board are presented in Figure 8a,b, respectively. The concentrator (designed by Will Whang and distributed as open-source [35]) contains two Semtech SX1257 front-end chipsets connected to a single Semtech SX1301 [36] baseband processor. The latter is also connected to the u-Blox MAX7Q GPS module, which provides the time reference via the Pulse-Per-Second (PPS) interface. The resolution of the timestamps of the GW is 1 μs (which corresponds to approximately 300 m distance traveled with the speed of light). The concentrator board was connected to the Raspberry Pi over the serial peripheral interface (SPI). The Raspberry Pi ran the Raspbian operating system with the open-source program lora_pkt_fwd [37] to process the data from the concentrator and communicate with the network server. Since it was a temporary measurement, each GW was powered from a 10 Ah power bank (with maximum 2 A current). Omnidirectional antennas with 1 dBi gain for 868 Mhz were used. In total, four boards have been produced and four GWs have been manufactured and used in the measurements.

### 5.2. End-Device

The TDoA-based localization done by the GWs does not require any hardware modifications for the end devices. However, to enable the measurement of the localization accuracy, a way to obtain the ground-truth position is needed. For this reason, specifically for these tests, we developed a test device integrating a LoRaWAN transceiver (i.e., RisingHF RHF76-052) and a GPS module (u-blox NEO-6M), see Figure 9. The designed test device, being controlled by a Nucleo F401 board, was configured to send a packet with a period of 30 s (operating with DR0, i.e., SF12). The coordinates obtained from the GPS module were encapsulated into every packet sent by the device and used as the ground-truth reference for our analysis. Note, that the LoRaWAN transceiver was configured as a device of class A. This was done for the two reasons. First, the class A functionality is obligatory to be supported by all LoRaWAN end devices and ensures minimum energy consumption of the device. Second, since the TDoA localization is based solely on the uplink transmissions, implementation of classes B and C does not bring any added value.

### 5.3. Experimental Setup Parameters Summary

Table 3 summarizes the key parameters for our experimental testbed.

## 6. Experiment Procedures and Pre-Testing

In this section, we start by reporting the results of the GW synchronization tests, which were conducted before the field measurements, and then detail our experimental procedures and the environment for the field tests.

### 6.1. Pre-Testing: Gateway Synchronization Validation

To validate the synchronization accuracy for the four manufactured LoRaWAN GWs, we carried out a special test. The measurements were performed in a rural environment in a field area with the coordinates of the site *49.379778${}^{{}^{\circ}}$ N, 15.961722${}^{{}^{\circ}}$ E*. The end-device was placed at this point, and the GWs were uniformly distributed in a circle with a 50 m diameter, centered at the end device and within its LoS. Note that for this measurement, the spreading factor used by the end-device was reduced to 8 (based on the results reported in [5], the choice of the spreading factor does not affect the accuracy of the localization).

Since the precise measurement of the transmission time by the end device is complex, we estimated and analyzed the difference of the timestamps of the GWs for the reception of the same uplink packet from the end device. The individual results (showing the worst-case scenario, i.e., the time difference between the maximum and the minimum timestamp) for 1000 test messages are illustrated in Figure 10. Both the mean and the median values equal 5 μs (corresponding to the ranging error of ≈ 1500 m). Note that these errors are the cumulative effect of several reasons, including the GPS errors, the GW internal clock instability, the NLoS propagation, etc.

### 6.2. Experiment Environment and Set-Up

The real-life tests were carried out in the countryside area outside of the urban area in the Vysočina region, Žďár nad Sázavou district, Czech Republic. Four GWs were used in the measurements—each placed in a village surrounding the village Závist (refer to Figure 6). The measurements were carried out using the 868 MHz band following the respective regulations of the EU region. The end device has been configured to send its packets with a period of 30 s using DR0 and 14 dBm transmit power. All the measurements were carried during one single day—the 29 March 2019. Note that the selected area was decently isolated and was located apart from the known position of the commercial LoRaWAN GWs. There were forests, fields, and village dwellings and no high-rise buildings in this area. The GWs were placed on elevated places to ensure proper communication conditions.

## 7. Measurement Results

In total, 131 messages were sent during our measurements, of which 95 messages were received by at least three GWs, and 61 messages (47%) were received by all four GWs. Based on the simulation results for this part of the study we picked up the best-performing algorithms. Specifically, for all the cases when the data from only three GWs were available, we used Chan’s algorithm. For the case when four GWs received a packet, we investigated the two other algorithms, namely Foy’s (iterative) and Chan’s (non-iterative). These two distinctive approaches are labeled on the following charts as “Chan (3GW)+Chan (4GW)” and “Chan (3GW)+Foy (4GW),” respectively. The results for the CDF are presented in Figure 15.

The mean localization error over all our measurements was ≈ 1.4 km, with the maximum error exceeding 13 km. Figure 11 illustrates the distribution of the error for the two tested algorithms (i.e., Chan (3GW) + Chan (4GW) and Chan (3GW) + Foy (4GW)).

Note that since only four GWs were employed in our measurements, we have not faced a problem of selecting the GWs, the measurements of which to be used; however, in a real-life multi-GW LoRaWAN network, this challenge will likely arise. Note that for the majority of the test locations, the error did not exceed 1.5 km. Figure 12 details the results further by illustrating the ground truth (blue cross) and the respective estimated (red cross) locations, as well as the positions of the GWs on the map of the area.

To improve the accuracy of the localization, we processed the results by using a moving median filter to smoothen the sharp transitions and suppress outliers. For this, we created two fixed-size first-in-first-out (FIFO) queues, as shown in Figure 13. The geolocation results were fed one by one into the first FIFO moving queue with a fixed size. From this queue, the median value was computed and further added to the second FIFO moving queue with a fixed size. The second queue’s purpose is averaging. Since the coordinates consist of two values (latitude and longitude), two independent FIFO queue systems were created for latitude and longitude. Note, that based on our experience, the suppression of outliers using median filtering (done by the first queue) had the most significant effect on reducing the position estimation error, compared to the averaging, done by the second queue.

The results, illustrating the mean/median error with filtering throughout all the experiments and the distribution of the errors, are presented in Table 4 and Figure 14, respectively. As can be seen, the use of filtering allowed to reduce the average error almost three times (from 1.5 km to 550–600 m), and the maximum error more than ten times (from over 13 km to about 1300 m). Note, that most of the time, the resulting error was within 500 m. Comparing the two algorithm sets tested, one can see that the use of the Chan + Foy combination allowed for reducing the average localization error by 56.5 m more than the use of the Chan algorithm standalone.

Figure 15 further shows the course of CDF when comparing the crude results to the results after filtration. Where the positive effect of filtration can be further observed. Figure 16 shows filtered geolocation results. It can also be seen that in some places, the results are more inaccurate than in other places. This is because in these places, it was not possible to have direct visibility to some of the GWs due to communication across the horizon, and thus NLoS communication occurred.

It should also be noted that the end-device tester was moving along the selected route within the measurement and was therefore not statically positioned. The larger the FIFO queue windows, the greater the delay in the result compared to the real position. Thus, it can be said that if the device is statically positioned, it would be possible to achieve higher geolocation accuracy.

## 8. Discussion

The results presented in this paper demonstrate the feasibility and shed some light on the achievable performance and the important aspects of localization based on the TDoA principles within the LoRaWAN LPWA networks. Despite being proven to be feasible even for low-cost implementation (as shown in this paper), the accuracy of such localization stays well below that of the GNSS and barely enables us to use this method for navigation applications; however, in the context of IoT, there are a number of applications for which the definition of just a region where an IoT device is located may be sufficient. Among these are the versatile applications to track goods, people, and even wildlife. Importantly, compared to the GNSS-based localization the TDoA approach has a number of attractive features.

The first one is related to the monetary cost. The TDoA-based localization by the GWs does not demand any hardware modifications or the addition of new hardware components for the end-devices. Meanwhile, the price of the basic GPS receiver chipsets as of today is in the order of 3–5 EUR in thousand-piece quantities. Second, the tested localization solution does not imply any additional energy consumption by the end device. To give an example, the low-power uBlox UBX-M8230-CT receiver in its lowest consuming Super-E (1 Hz) mode consumes 8 mW [38]. Third, the proposed method does not introduce any communication overhead and does not need any extra time to get locked. All in all, based on these facts and despite being not precisely accurate, we consider the TDoA localization to be a perspective solution for a number of applications, including, e.g., transportation, goods, and wildlife tracking. Another two undoubted advantages of TDoA localization are (i) that it can be applied to devices already deployed in the field with no GNSS receiver on board, (ii) that it can be potentially used also indoors, where no satellite coverage is available.

The simulations, conducted as a part of our pre-study, showed that Chan’s and Foy’s algorithms can be effectively used for LoRaWAN localization for the cases when the signal is received by three and four GWs, respectively. The combination of the two algorithms (i.e., using the results of Chan’s algorithm as the initial estimate for Foy’s algorithm) provided the best localization accuracy in our case. Introducing the post-filtering and averaging during the practical measurements, we managed to define the position for about half of the test cases with the mean error of 543 m (median error of 424 m). These results are comparable with the ones reported in [5], where six GWs (notably, of V2) covered the area of 0.43 km${}^{2}$ reporting the geolocation accuracy in between 20–200 m (note, that the details of the software algorithms used in these work are not reported). The authors of this study achieve higher localization accuracy due to the smaller test area and the higher probability of LoS communication due to more GWs used, and using gateways with a precise clock (timestamp resolution 1 ns). It should be noted that our study used a gateway designed according to the reference design (V1), which does provide lower resolution timestamps (1 μs) for the incoming messages. The main difference is in the price—the new gateways with precise timestamping feature cost about $1400, while the price of basic V1 gateways starts from $150. Another important aspect, which raises the importance of enabling the TDoA localization for V1 GWs is that over the past five years while LoRAWAN is being deployed a great number of V1 GWs were already installed and are now in action. Meanwhile, deployment of an infrastructure based on V2 GWs would imply substantial investments, which may not be tolerable for all the operators.

It can be observed that even the best optimization is not able to overcome the physics of both signal processing and multipath propagation, which greatly affects the geolocation precision. Still, we consider that the accuracy of TDoA localization over LoRaWAN can be further refined using machine learning and other statistical methods, such as the tracking improvement algorithm [23].

Another perspective approach to improve the localization accuracy is the increase of the timestamp resolution and accuracy. For this, the design of the LoRa radios and the GW needs to be re-thought, introducing more accurate clocks and FPGA (these GWs are referred to as “V2” GWs, in contrast to V1, which are being mostly deployed in the LoRaWANs of today). However, all these modifications increase the price of the GWs. Also, the possibility to increase the bandwidth of the signal used for the localization should be investigated. However, the latter may require modification of the LoRaWAN regional recommendations (at least for the EU region).

## 9. Conclusions

In the current paper, we comprehensively discussed the use of TDoA-based localization in LoRaWAN networks. We started by discussing the background and the alternative localization approaches possible for LoRaWAN, and then focused on the specifics of the TDoA-based localization. Using simulations, we studied the potential performance of five classic TDoA algorithms, namely: Fang’s, Chan’s, SI, LLS, and Foy’s algorithms. The two algorithms providing the best localization accuracy, namely Chan’s algorithm for two TDoAs (i.e., data from 3 GWs) and Foy’s algorithm for three TDoAs (i.e., data from 4 GWs) were selected for the further validation through the field measurements.

For the field measurements, we created four low-cost Raspberry PI based LoRaWAN GWs equipped with a GPS receiver for time synchronization capable of TDoA localization and a special test device equipped with a GNSS signal receiver. During the first stage of our experiments, we estimated the clock accuracy of the GWs, which was shown not to exceed 5 μs. During the second stage of the experiments, which were conducted in a countryside area in the Vysočina region of the Czech Republic, we have estimated the accuracy of the localization.

Of the 131 experiments conducted, for about 47% of the cases, the localization by the two selected algorithms converged. The mean localization error over all our measurements was ≈ 1.4 km, with the maximum error exceeding 13 km. To improve these results, we tried using a moving median filter to smoothen the sharp transitions and suppress outliers. This allowed us to reduce the average error almost three times to 550–600 m (median error 424 m), and the maximum error more than ten times (to about 1300 m). The use of the Chan + Foy algorithm combination allowed reducing the average localization error by 56.5 m compared to the use of the Chan algorithm standalone. Simultaneously, as we have shown in this paper, the repetition of the experiments and post-filtering may improve the accuracy of localization even further. However, note that the presented in this paper results and conclusions are based on just one day of measurements in one specific location and thus may not have captured all phenomena, and require a further validation through a more extensive (in space and time) measurement campaign.

All in all, our results demonstrate that the TDoA-based localization is a valuable addition to the basic connectivity service provided by the network. Even though the accuracy of this approach as of today is much lower than that of the conventional GNSS, it does not impose any cost and energy overheads for the end devices, and only very minor overheads (i.e., the costs of a GNSS receiver used for time synchronization) for a LoRaWAN GW. Among the real-life applications, benefiting from such functionality are transportation, goods, and wildlife tracking, to name just a few. Another potentially perspective direction is to use this method for the use cases when GNSS receiver cannot be used due to energy or size constraints (e.g., in tiny mobile tags) and for areas located outside of the satellite reach (e.g., indoors, tunnels, and mines).

To advance the usability and performance of TDoA-based localization in LoRaWAN, started by this study, the efforts along a number of tracks are needed. First, there is a need for the hardware improvement of the GWs to enable more accurate timestamping and synchronization between these. Second, the effect of the LoRa signal parameters (e.g., the spreading factor, the coding rate, the bandwidth, and the channel) on the accuracy of localization should be researched. Third, the more advanced algorithms for localization (e.g., to select the GWs, measurements to be used, detect LoS vs. NLoS, and enable mapping-based localization) should be considered. Forth, more extensive and spatially distributed measurements are needed to understand the temporal fluctuations and the effects of the environment and its changes.

## Figures and Tables

**Figure 1 sensors-20-05464-f001:**
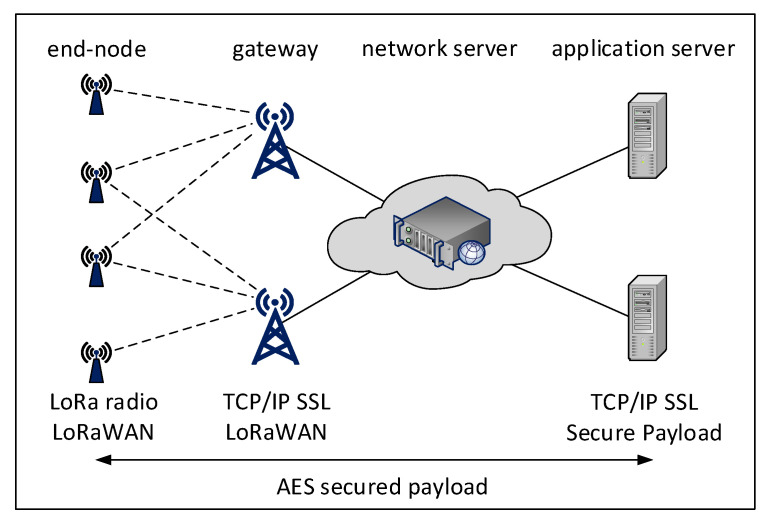
General long-range wide-area-network (LoRaWAN) architecture with end-to-end security using advanced encryption standard (AES) and secure sockets layer (SSL) for TCP/IP connections.

**Figure 2 sensors-20-05464-f002:**
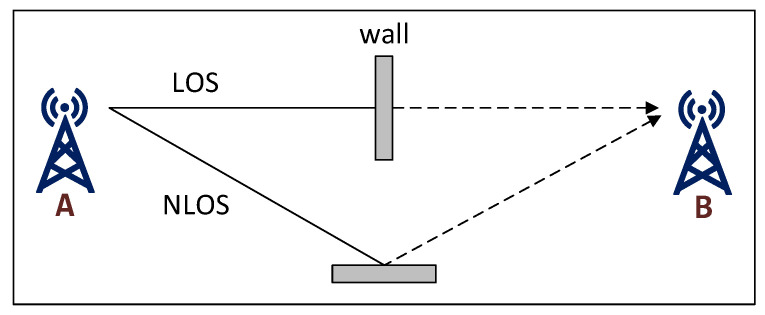
Line-of-sight (LoS) vs. non-line-of-sight (NLoS) signal propagation.

**Figure 3 sensors-20-05464-f003:**
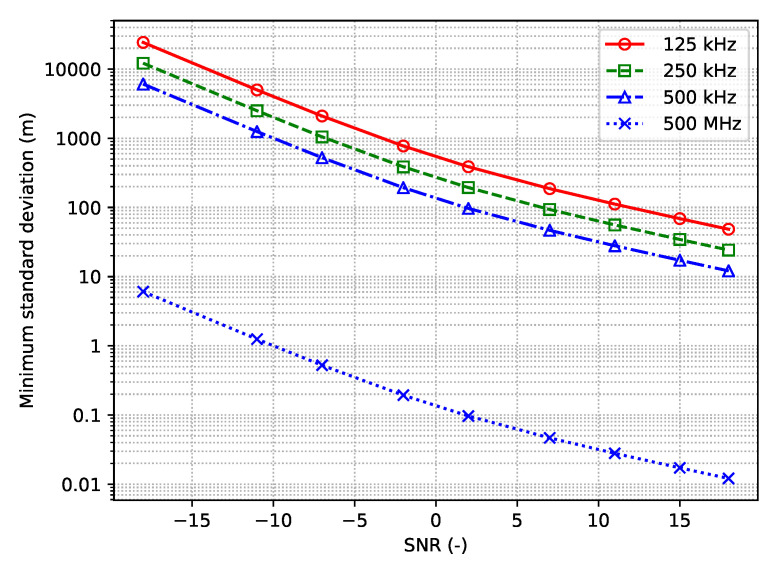
The Cramer–Rao bound for selected signal bandwidths.

**Figure 4 sensors-20-05464-f004:**
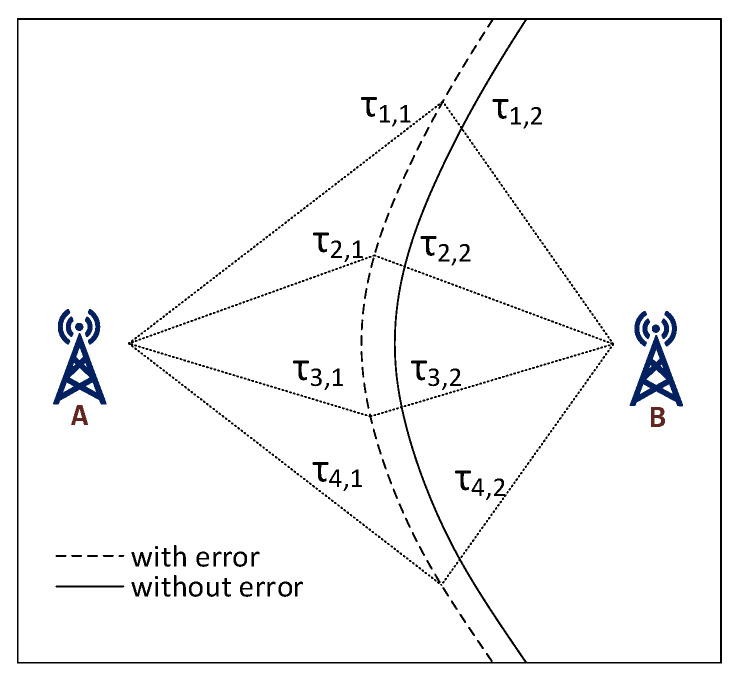
Time difference of arrival (TDoA)—hyperbolic trajectory principle.

**Figure 5 sensors-20-05464-f005:**
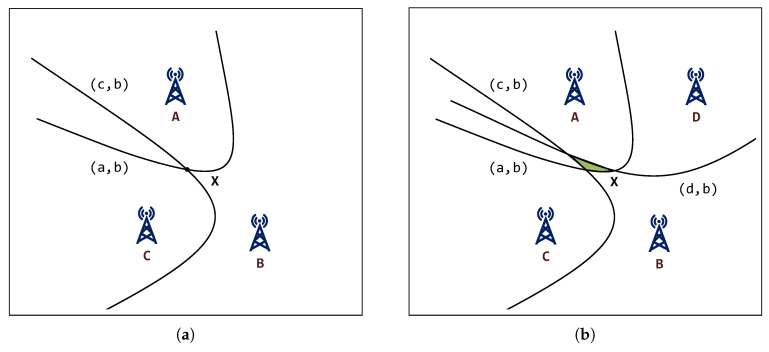
Illustration of a position estimation using TDoA. (**a**) TDoA—3 receivers. (**b**) TDoA—4 receivers.

**Figure 6 sensors-20-05464-f006:**
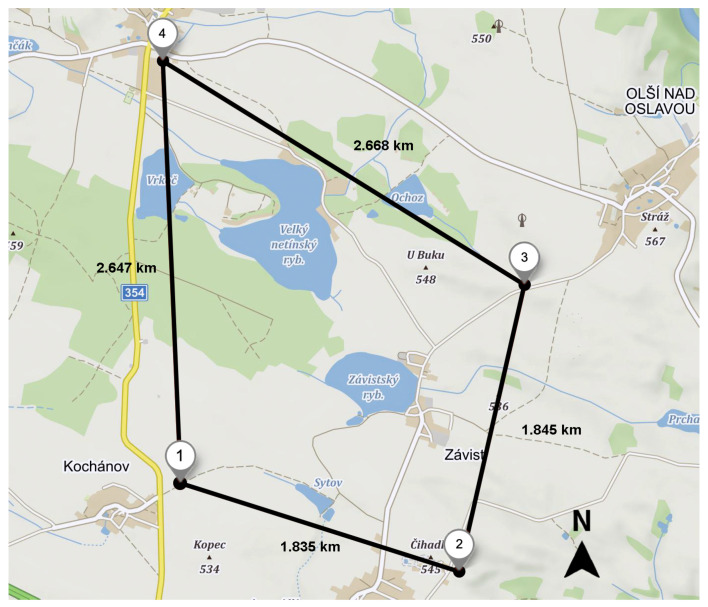
Selected positions of the gateways (GWs) used for simulations and empirical measurements.

**Figure 7 sensors-20-05464-f007:**
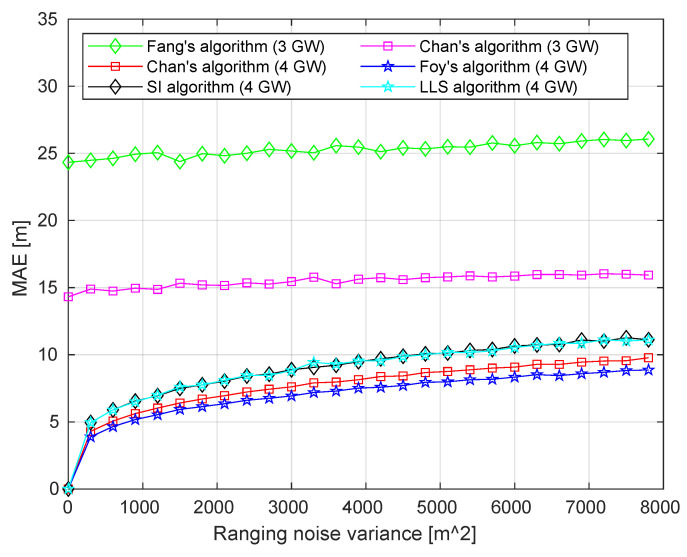
Ranging noise vs. mean absolute error (MAE) for selected TDoA algorithms.

**Figure 8 sensors-20-05464-f008:**
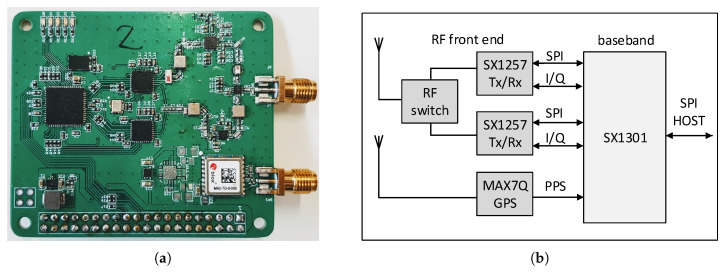
LoRaWAN concentrator (SX1301 + 2x SX1257 + GPS). (**a**) Concentrator board. (**b**) Block diagram of a concentrator [35].

**Figure 9 sensors-20-05464-f009:**
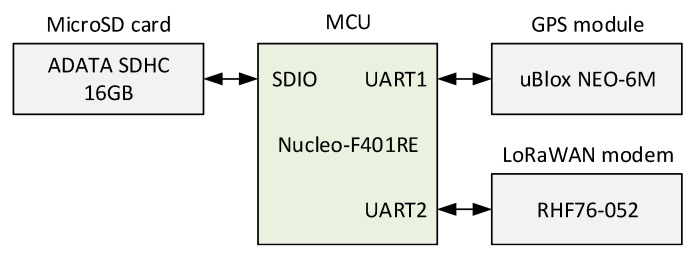
Block diagram of the end-device used for testing.

**Figure 10 sensors-20-05464-f010:**
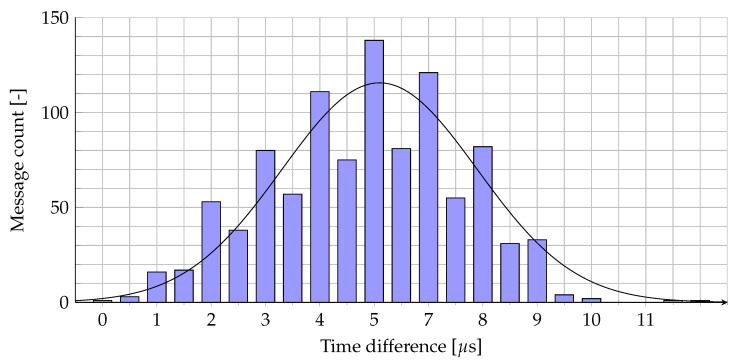
Timestamp fluctuations between 4 GWs.

**Figure 11 sensors-20-05464-f011:**
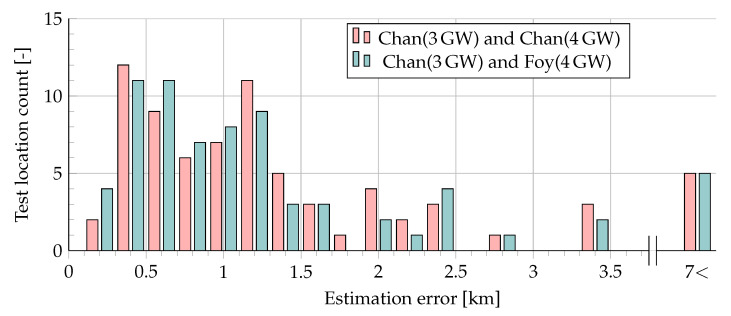
Comparison of raw estimation results (3 and 4 GWs) using Chan’s and Foy’s algorithms.

**Figure 12 sensors-20-05464-f012:**
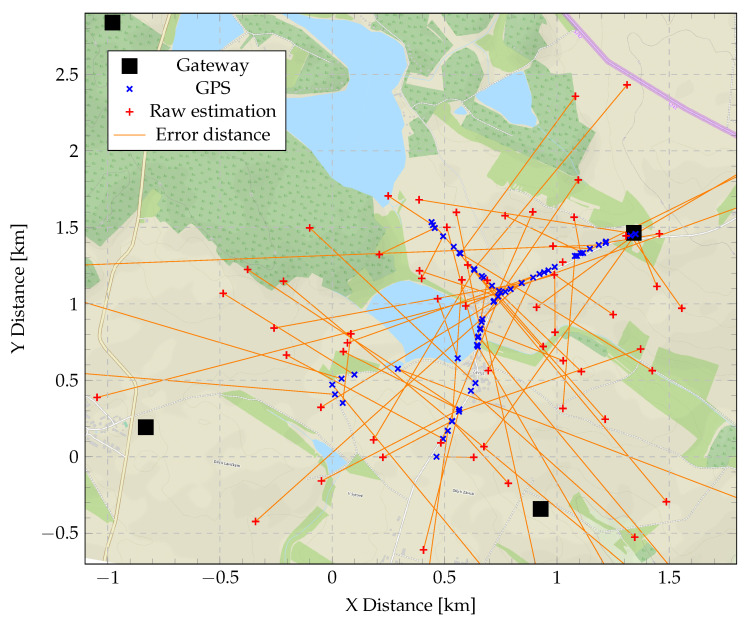
Position estimation (Chan’s and Foy’s algorithms) compared to the correct GPS position.

**Figure 13 sensors-20-05464-f013:**
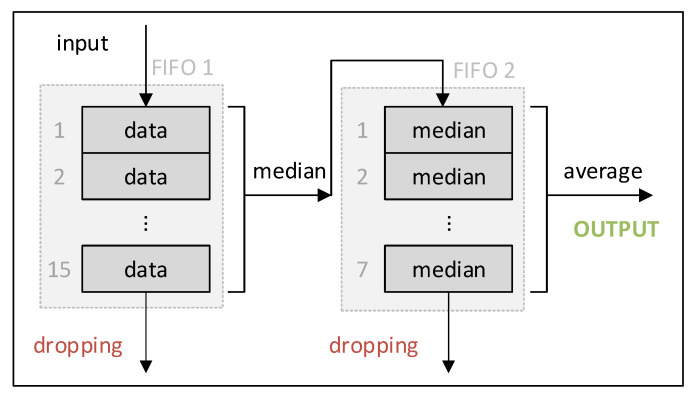
Procedure of filtering by averaging of moving medians.

**Figure 14 sensors-20-05464-f014:**
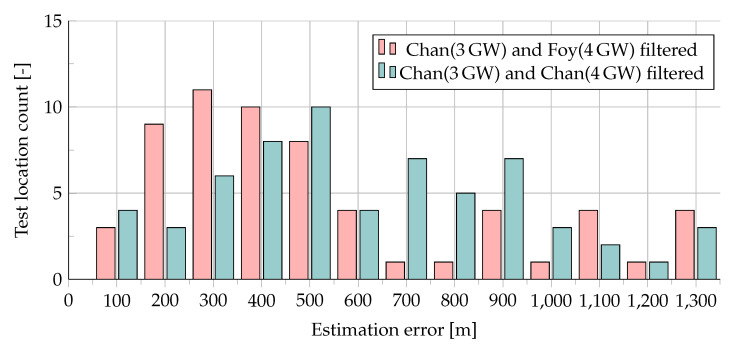
Filtered Chan’s and Foy’s algorithms (queues - median: 10, average: 6).

**Figure 15 sensors-20-05464-f015:**
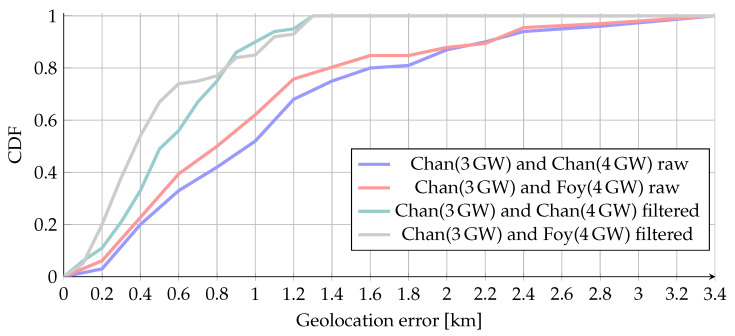
CDF of estimation results (3 and 4 GW) using combination of Chan’s and Foy’s algorithms before and after filtration.

**Figure 16 sensors-20-05464-f016:**
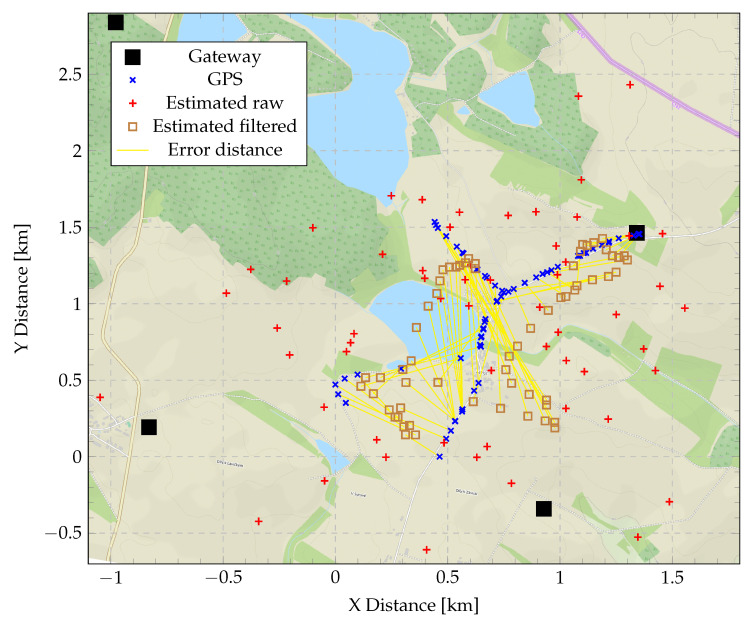
Position estimation using Chan’s and Foy’s algorithms after filtration compared to the real GPS position.

**Table 1 sensors-20-05464-t001:** Comparison of experiment setup and the achieved results in the related works.

Est. Error [m]	GW Number	Test Area	Approach	Reference
**TDoA (outdoor)**				
20–200 (avg.)	6	0.65 × 0.65 km	-	[5]
100 m (avg.)	4	1.74 × 1.74 km	-	[13]
200 (med.)	19	10 × 10 km	Maximum Likehood Estimation (MLE)	[16]
500 (med.)	-	10 × 10 km	Multilateral Dissection (MLD)	[16]
**RSSI (indoor)**				
20–30 (avg.)	3	180 × 80 m	Minimum Mean Square (MMS)	[17]
**RSSI (outdoor)**				
10 (avg.)	6	150 × 100 m	Weighted Least Squares (WLS)	[18]
18 (avg.)	3	150 × 100 m	Weighted Least Squares (WLS)	[18]
9–20 (avg.)	3	120 × 200 m	-	[19]
24 (avg.)	4	340 × 340 m	Fingerprint algorithm	[20]
398 (avg.)	68	7.28 × 7.28 km	Fingerprint algorithm	[21]
357 (avg.)	68	7.28 × 7.28 km	Neural network	[22]

**Table 2 sensors-20-05464-t002:** Computation time comparison between Fang, Chan, linear least-squares (LLS), spherical-intersection (SI), and Foy algorithms [34].

	Fang	Chan	LLS	SI	Foy
3 GWs	598.49 μs	717.95 μs	-	-	-
4 GWs	-	1 400.80 μs	1 012.00 μs	951.83 μs	4 705.20 μs

**Table 3 sensors-20-05464-t003:** Experimental testbed parameters.

**End-device**	
LoRaWAN device class	A
Data rate (DR)	0
Send period	30 s
Transmitt power	14 dBm
Frequency	868 MHz
LoRaWAN module	RisingHF RHF76-052
GPS	u-Blox NEO-6M
MCU	STM32F401
End-device count	1
**Gateway**	
Frequency	868 MHz
Timestamp resolution	1 μs
GPS	u-BLox MAX7Q
Concentrator	Custom PCB (SX1301 + 2xSX1257)
Computer	Raspberry Pi 3B+
Power bank	10 Ah
Gateway count	4

**Table 4 sensors-20-05464-t004:** Comparison of precision using different combinations of geolocation algorithms with post-filtering.

	Chan (3 and 4 GW)	Chan (3 GW) + Foy (4 GW)
Median FIFO size	10	10
Averaging FIFO size	6	6
Averaging median filtering (mean error)	593.77 m	542.97 m
Averaging median filtering (median error)	544.78 m	423.79 m

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
