# Peer review of "Investigation of the Performance of TDoA-Based Localization Over LoRaWAN in Theory and Practice"

_sensors, 2020, doi:10.3390/s20195464_

Round 1

Reviewer 1 Report

This paper comprehensively discussed several state-of-art TDoA-based localization in LoRaWAN networks. Furthermore, it studied the theoretical performance of five classical TDoA algorithms (Fang’s, Chan’s, SI, LLS and Foy’s ones) by simulations. The two algorithms providing the best localization accuracy namely Chan’s algorithm and Foy’s algorithm had been selected for the further validation via the field measurements. According to the results provided in the work, the TDoA-based localization is a valuable addition to the basic connectivity service provided by the network without imposing any cost and energy overheads for the end devices.

The following comments/ questions shall be addressed:

  1. Line 152-153, where is the plot?

  1. Line 191, what are the selection criteria for the four accessible locations?

  1. In 4.4, “the Foy’s algorithm for 3 GWs” is contradictory to “For the other algorithms (i.e., SI, LLS, and Foy’s), the TDoAs from at least four GWs need to be used.” in line 216.

  1. Line 250, what is the class B and class C? Why did you use the devices of class A in this work?

  1. Line 348, no error distribution of Chan (4GW) + Foy (4G) in Figure 10?

This work exams the selected algorithms in simulation and field tests. Although its not a very interesting work, the contents are solid, and the results are convincing. Thus, the reviewer suggests to accept the work after the above concerns have been handled.

Reviewer 2 Report

The paper is well written and it's also pleasant to read (a check of the language should be done anyway), but it has flaws such that it can not be accepted in its current form.

The literature is insufficient and needs to be expanded with more references to the state of the art of WAN localization and the various techniques.
In addition, a few more references to similar localization systems should be included in the comparison.

The way the algorithms are applied during the experimental tests is not very clear and there is a lack of CDF graphs which are important to evaluate localization errors.

The number of points tested is not particularly high, only 61 actual points?
Are all points distinct? It would be useful to test the device at some fixed positions and acquire several packet at the same point and evaluate the cumulative error.

The best algorithms may depend on the specific setup and operating and environmental conditions and this has not been considered at all.

The case study of the experiments is quite limited. The tests were performed in a rural area, while in a city environment the results should be even worse.
The LoRa alliance whitepaper presents tests conducted in a port area with several reflections (metallic buildings) and still presents better results (about 90% of errors are below 70m and without filtering).

Authors should investigate more closely the causes of errors in their setup, and consider how to reduce them.

The conclusions and the discussion part should be partly reviewed, the results are not particularly positive and make the system, as presented, unusable except to identify a macro region where the device is located. However, navigation would be impossible with the results of this paper.
A system with a GPS chip to be used with minimal activation times would still give much better results, especially in rural areas without obstacles, and turning on the localization system from time to time, at rates higher than the LORA trasmission, would not significantly increase consumption.

Reviewer 3 Report

In this paper, the authors compared the performance of the five classical TDoA based localization algorithms. It can give useful information about TDoA-based localization in LoRaWANs to the readers; however, it has some issues that need to be fixed as follows:

It is better to add a general figure of LoRaWAN networks including its members to the introduction section.

In addition to the reference number, please include the names of comparing approaches in Table 1.

Add another table to Section 2.1 comparing the proposed methods based on the qualitative features like energy, delay and so on. The related work section should be extended by explaining these details.

The algorithms of Fang’s [20], Chan’s [21], Spherical-Intersection (SI) [22], Linear-Least Squares (LLS) [23], and Foy’s [24] should be briefly explained and qualitatively compared.

The experimental setup parameters should be summarised within a table.

Round 2

Reviewer 2 Report

The paper is improved and can be considered for approval.

Reviewer 3 Report

Thanks to the authors who revised the manuscript. It can be accepted in the current format.